# Enhanced Pharmaceutically Active Compounds Productivity from *Streptomyces* SUK 25: Optimization, Characterization, Mechanism and Techno-Economic Analysis

**DOI:** 10.3390/molecules26092510

**Published:** 2021-04-25

**Authors:** Muhanna Mohammed Al-Shaibani, Radin Maya Saphira Radin Mohamed, Noraziah Mohamad Zin, Adel Al-Gheethi, Mohammed Al-Sahari, Hesham Ali El Enshasy

**Affiliations:** 1Micro-Pollutant Research Centre (MPRC), Faculty of Civil Engineering & Built Environment, Universiti Tun Hussein Onn Malaysia, Parit Raja, 86400 Batu Pahat, Malaysia; muhanan@uthm.edu.my (M.M.A.-S.); mohammedalsahari@gmail.com (M.A.-S.); 2Center for Diagnostic, Therapeutic and Investigative Studies, Faculty of Health Sciences, Universiti Kebangsaan Malaysia, Jalan Raja Muda Abdul Aziz, 50300 Kuala Lumpur, Malaysia; 3Institute of Bioproducts Development (IBD), Universiti Teknologi Malaysia (UTM), 81310 Skudai, Malaysia; henshasy@ibd.utm.my; 4City of Scientific Research and Technology Applications (SRTA), New Burg Al Arab, 21934 Alexandria, Egypt

**Keywords:** pharmaceutical active compounds, fermentation, optimization, *Streptomyces* SUK 25, response surface methodology, techno-economic analysis

## Abstract

The present research aimed to enhance the pharmaceutically active compounds’ (PhACs’) productivity from *Streptomyces* SUK 25 in submerged fermentation using response surface methodology (RSM) as a tool for optimization. Besides, the characteristics and mechanism of PhACs against methicillin-resistant *Staphylococcus aureus* were determined. Further, the techno-economic analysis of PhACs production was estimated. The independent factors include the following: incubation time, pH, temperature, shaker rotation speed, the concentration of glucose, mannitol, and asparagine, although the responses were the dry weight of crude extracts, minimum inhibitory concentration, and inhibition zone and were determined by RSM. The PhACs were characterized using GC-MS and FTIR, while the mechanism of action was determined using gene ontology extracted from DNA microarray data. The results revealed that the best operating parameters for the dry mass crude extracts production were 8.20 mg/L, the minimum inhibitory concentrations (MIC) value was 8.00 µg/mL, and an inhibition zone of 17.60 mm was determined after 12 days, pH 7, temperature 28 °C, shaker rotation speed 120 rpm, 1 g glucose /L, 3 g mannitol/L, and 0.5 g asparagine/L with R^2^ coefficient value of 0.70. The GC-MS and FTIR spectra confirmed the presence of 21 PhACs, and several functional groups were detected. The gene ontology revealed that 485 genes were upregulated and nine genes were downregulated. The specific and annual operation cost of the production of PhACs was U.S. Dollar (U.S.D) 48.61 per 100 mg compared to U.S.D 164.3/100 mg of the market price, indicating that it is economically cheaper than that at the market price.

## 1. Introduction

Methicillin-resistant *Staphylococcus aureus* (MRSA) strain is a group of strains belonging to *S. aureus* that has acquired resistance to a class of beta-lactam antibiotics. This strain represents a worldwide health problem due to its capability to become resistant to the currently available antibiotics. Moreover, MRSA infections can become severe and cause sepsis and life-threatening disease [1]. *Streptomyces* species have been isolated from a different environment with high potential to produce more than 10,000 of the pharmaceutically active compounds (PhACs), including alkaloid, polyene macrolide, saccharide, and antibiotics [2]. However, the PhACs produced depend on the source of *Streptomyces* species. Therefore, more studies are required to explore the ability of *Streptomyces* sp. to produce PhACs with high antimicrobial activity against pathogenic microorganisms. Moreover, production efficiency relies on external factors including temperature, pH, incubation time, and shaker rotation speed, and several internal factors such as the sources of nitrogen and carbon, which should be optimized. 

The one-variable-at-a-time approach design is time-consuming and complex to conduct for each separated factor, since the optimization method entails the use of a single parameter for every trial [3,4]. For the optimum levels to be ascertained, multiple numbers of experimental trials need to be conducted. One of the investigational techniques is the response surface methodology (RSM), which represents an important tool that employs mathematical models and statistics for the optimization of parameters of fermentation processes. Besides, RSM consists of multivariable polynomial models, which are used to optimize a response based on a given set of variables. In addition, there is a wide application of the RSM approach in the optimization of microbial fermentation processes, as well as in the determination of the effect of various factors. A different approach used to increase engineered strain’s expression is the application of biostatistics to optimize culture conditions, with the RSM optimization with central composite design (CCD) being the widely accepted design [5]. 

The gas chromatography mass spectroscopy (GC-MS) analysis involves the combination of technologies for the analysis of chemical compounds. The GC executes the separation of the compounds, whereas the MS produces the specific mass profiles of each of the detected compounds. The analysis of emissions from biogenic volatile organic compounds (VOCs) is commonly executed using GC-MS [6]. 

VOCs are typically small and odorous compounds manufactured as secondary metabolites produced by some microorganisms and microorganisms associated with plants and soil [7]. 

The chloramphenicol (CAP) and seven diketopiperazines (DKP) were isolated, purified, and identified from SUK 25 as a bacteriostatic antibiotic [3,4]. The mode of action of CAP and DKP involves the inhibition of protein synthesis, which is achieved by preventing the elongation of the protein chain through the inactivation of the activity of peptidyl transferase present in the bacterial ribosomes as reported by Zin et al. [8]. 

Gene ontology (GO) is a bioinformatics technique that combines intra- and inter-species representation of gene and gene product attributes. GO consists of three main aspects: the biological process, its molecular function, and the cellular function, as the technique allows operators to describe and define a gene/gene product in detail [9]. 

The techno-economic analysis (TEA) is an ingrained process, which is developed in performance with technology, to confirm that market-driven prices can be realized [10]. TEA is a part of the stage-gate process in product development and related research. It is a technically and economically interrelated invention and encouragement in most industries [11]. TEA is a study of a production process in an industry sector to determine roughly how effectively the economy or something within it is operating. In addition, it assesses the details of how you intend to deliver a product or service to customers and has shown the costs involved in the production of products [12,13]. 

The present study aimed to optimize the PhACs productivity from *Streptomyces* SUK 25 in submerged fermentation (SmF) using RSM. Furthermore, the characteristics and mechanism of action of the PhACs against MRSA were determined. Besides, the techno-economic analysis of PhACs production was estimated to studying the applicability of the generated compounds to be used as an alternative for the commercial antimicrobial products.

## 2. Results

### 2.1. RSM-Based Optimization of PHACs Production

The optimization of the production using SmF process was conducted with seven independent factors and three dependent variables as represented in Appendix A. The maximum crude extracts production ($y_{1})\;$was 8.20 vs. 8.58 mg/L of the actual and the predicted results, respectively. The maximum MIC ($y_{2})\;$activity obtained 10.40 vs. 8.00 µg/mL for the predicted and actual results, respectively. For the maximum I.Z ($y_{3})$ values, the predicted and actual results were 18.03 vs. 17.60 mm, respectively. These findings were recorded after 12 days, at pH 7, temperature 28 °C, speed 120 rpm, 1 g glucose /L, 3 g mannitol/L, and 0.5 g asparagine/L. These results specified that the explored factors represent an essential starring role in the production of crude extracts, increasing inhibition zone (I.Z), and improving the MIC activity. The productivity of the amount of crude extracts ($y_{1}$) was associated positively and significantly (*p* < 0.05) with the incubation time (days)$\;x_{1}$ and pH $\;x_{2}\;\mathrm{factors}$, while having a non-significant correlation with other factors $x_{3}-x_{7}$, as described in Table 1 and Appendix A. In contrast, the incubation time $x_{1}$, pH $x_{2}$, and temperature $x_{3}$ revealed a significant negative quadratic effect on amount of crude extract ($y_{1}$), while the other factors $x_{4}-x_{7}$ had a non-significant quadratic effect on the amount of crude extracts. The good activity and low MIC ($y_{2}$) was related positively and statistically significant (*p* < 0.05) with factors$\;x_{3},\;x_{4,\;}x_{5},\;\mathrm{and}\;x_{7}$, while having a non-significant correlation with factors$\;x_{1}\;\mathrm{and}\;x_{2}$. In contrast, all responses exhibited a significant negative quadratic effect on the MIC values. The best and wide I.Z ($y_{3}$) was related positively and statistically significant (*p* < 0.05) with factors$\;x_{1},\;\mathrm{and}\;x_{2,\;}$, while having a non-significant correlation with factors $\mathrm{temperature}\;x_{3},\;\mathrm{speed}\;x_{4,\;}\;\mathrm{glucose}\;x_{5,\;}\;\mathrm{mannitol}\;x_{6},\mathrm{and}\;\mathrm{asparagine}\;x_{7}$. In contrast, $x_{1},\;\;x_{2},\;x_{3},\;\mathrm{and}\;x_{4}$ displayed a significant negative quadratic effect on the size of the I.Z, while there was a non-significant correlation$\;\mathrm{for}\;x_{5,\;}x_{6}\;,\mathrm{and}\;x_{7}$ factors, as described in Table 1. The ANOVA analysis for the quadratic model, as clarified in Table 2 and Appendix A, indicated that independent factors ($x_{1}-x_{7}$) contributed significantly (*p* < 0.05) with R^2^ = 0.70 of the coefficient for amount of $\mathrm{crude}\;\mathrm{extracts}\;y_{1}$, R^2^ = 0.63 for MIC $y_{2}$, and R^2^ = 0.75 for the I.Z $\;y_{3}$. These findings verify the appropriateness of the model.

The standard error of regression was used to detect the fitting of the experimental and predicted results. The value of the standard error of regression recorded in this study ranged from 0.49 to 2.17 for amount of crude extracts ($y_{1})\;$and from 0.79 to 3.52 for I.Z$\;y_{3}$, which indicate the accuracy of the experimental results. In contrast, the standard error of $y_{2}$ was between 8.22 and 24.63; this error was high and might be related to the nature of the response and measurement methods where the MIC represented good response when it was in low value. 

The equation for the linear and quadratic influence of the independent factors on the dependent variables were presented in Equations (1)–(3).
(1)$$y_{1}=1.86+2.49\;x_{1*}+3.07x_{2*}+0.22x_{3}-1.13x_{4}-0.39-0.099x_{6}-0.27x_{7}-4.42x_{1*}^{2}-2.19x_{3}^{2}+0.22x_{4}^{2}+1.36x_{5}^{2}+1.70x_{6}^{2}+1.31x_{7}^{2}\;$$
(2)$$y_{2=}10.03+11.90\;x_{1}+1.79x_{2}-17.01x_{3*}-25.60x_{4*}--26.50x_{5*}+13.41x_{6*}+8.03x_{7}-8.48\;x_{1}^{2}-0.015x_{2}^{2}-33.44x_{3*}^{2}-24.00x_{4*}^{2}-26.15\;x_{5*}^{2}-10.75x_{6}^{2}\;19.03x_{7}^{2}$$
(3)$$y_{3}=9.07+6.48x_{1*}+4.18x_{2*}-0.75x_{3}-1.93x_{4}-1.45x_{5}+0.63x_{6}-0.25x_{7}-7.23x_{1*}^{2}-1.68x_{2}^{2}-3.37x_{3}^{2}-1.61x_{4}^{2}+1.33x_{5}^{2}+2.30x_{6}^{2}+1.59x_{7}^{2}\;$$
where * represents the factors that have a significant role; $x_{1}$ (time) (day),$\;x_{2}$ (pH), $x_{3}$ (temperature) (°C),$\;x_{4}$ (speed) (rpm), $x_{5}$ (glucose) (g/L), $x_{6}$ (mannitol) (g/L), $x_{7}$ (asparagine) (g/L), $y_{1}$ (crude extracts) (mg/L), $y_{2}$ (MIC) (µg/mL),$\;y_{3}$ (I.Z) (mm).

### 2.2. Validation of the Optimal Parameters

The best operating parameters for producing PhACs were determined after 12 days at pH 7, temperature 28 °C, shaker rotation speed 120 rpm, (1 g) glucose /L, (3 g) mannitol /L, (0.5 g) asparagine/L with 95% of the confidence level and 0.5% of significance. The autonomous considerations correlated significantly (*p* < 0.05) just at perfect circumstances of the SmF process. This evidently shows that interaction effects occur, as one factor’s influence depends mostly on another factor as represented in (Figure 1a–l). The results revealed that the internal factors glucose, mannitol, and asparagine ($x_{5},\;x_{6},\;x_{7})$ have independent effects on the productivity of amount of crude extract $y_{1}$ without a significant interaction (Figure 1a–c). However, the external factors $x_{1}$ (incubation time),$\;x_{2}$ (pH), $x_{3}$ (temperature),$\;x_{4}$ (speed) exhibited a significant synergistic effect on the productivity of the amount of the crude extract as represented in (Figure 1d,g,k,l). Moreover, the external factors showed more influence on the amount of crude extract $y_{1}$ compared to the internal factors. Nonetheless, the external factors exhibited a synergistic interaction with the internal factors (Figure 1e,f,h–j)). In contrast, the data analysis of the interactions of the factors revealed that the internal factors ($x_{5},\;x_{6},\;x_{7})$ have a synergistic interaction to MIC activity as shown in (Appendix A). The internal factors exhibited significant synergistic effects on MIC activity, as revealed in (Appendix A), more than the external factors (Appendix A–l)). The internal factors have no significant interaction and effects on$\;y_{3}$ (Appendix A), while the external factors exhibited more efficiency in improving $y_{3}$ (Appendix A–c,g,h). 

The Plackett–Burman design was used to screen and select the components of the compelling media. The resulting data indicated that the internal factors such as glucose, asparagine, and mannitol had positive impacts in contrast to the other components in the medium. Additionally, other external factors including the pH, duration of incubation, and shaker rotation speed had positive effects. In this study, the obtained R^2^ value (0.70) indicated that the model explained 70% of the overall variation. When R^2^ is closer to 1, the strength of the model is improved and makes a better prediction of the response. Following the RSM-based optimization, the increment in the antibacterial activity of PHACs was affirmed after comparing the results to the one-at-a-time strategy design medium. With deference to the weight of the crude extract, the antibacterial activities against MRSA ATCC 43300 and MIC were improved from 5.6 to 8.2 g/L (42.9%) and from 16 to 8 and 4 µg/mL (50%), respectively. 

The increment in the zone of inhibition was from 11.4 ± 1.5 to 17.2 ± 1.5 mm. Based on the (50.9%) increase in the modified media, the finding suggests that the antibacterial metabolite production by SUK 25 was affected by the quantity of the media components. Therefore, the present experimental design enhanced the optimization of significant media components with a high level of accuracy. Accordingly, this is the first study to report a 50.9% increase in antibacterial activity against MRSA ATCC 43300 from an endophytic *Streptomyces* SUK 25.

### 2.3. Column Chromatography and Thin-Layer Chromatography 

The present study showed that 13 fractions were collected from the crude ethyl acetate extract of SUK 25 according to the gradient elution solvent by using column chromatography. The fractions with similar R*_f_* in TLC were evenly mixed. Only seven pure compounds were separated from fraction number 2 and fraction number 7. These compounds were purified, identified, and characterized by using HPLC, LS-MS, and NMR as reported by our previous studies carried by [6,7].

### 2.4. Determination of Antibacterial Activity by Using MIC and Disc Diffusion Method 

The mean MICs of the triplicate samples for fraction number 2 and fraction number 7 at the best fermentation conditions against MRSA ATCC 43300 were 4 and 8 µg, respectively. In addition, the mean diameters of the triplicate samples for I.Z against the same bacteria were 17.2 ± 1.2 mm and 17 ± 1.3 mm. 

### 2.5. Fourier Transform Infrared Spectroscopy 

The presence of different compounds from various functional groups was detected. In this study, the FTIR spectroscopy demonstrated its reliability and sensitivity for the detection of bimolecular composition. The dominant bands in the case of crude extract were observed at 3348, 3319, 2945, 2833, 1558, 1449, 1394 cm^−1^. The bands at 1121, 1092, and 972 cm^−1^ are due to the presence of C–O stretch (primary alcohol) and =C–H bend alkenes, O–H bending as carboxylic acids. The absorption spectra of the column compounds’ extract samples are depicted in (Figure 2) and (Appendix A). 

### 2.6. GC-MS Analysis of Volatile Components 

The GC-MS analysis of the volatile components of *Streptomyces* SUK 25 using the NCBI PubChem bioassay database. The peaks were prudently identified based by comparing the features of the mass spectral and NIST database. The analysis revealed that among the 38 peaks, as demonstrated in (Figure 3), the 21 compounds that demonstrated the presence of bioactive constituents have been previously documented for their antioxidant, antimicrobial, antifungal, and anti-adherence activities. Other functions include neurotropic action and presence of anti-inflammatory activates such as n-dodecane (1), eicosane (2), phenol, 2,5-bis (1,1-dimethyl ethyl) (3), cetene (4), diethylphthalate (5), 2-methyloctacosane (6), 1-octadecane (7), phthalic acid, isobutyl nonyl ester (8), heneicosane (9), and n-hexa decanoic acid (10) were isolated from fraction no. 2 as represented in Table 2. From fraction no. 7, eleven compounds were identified as di-butyl phthalate (11), 1-nonadecene, thieno [3,2-e] benzofuran(12), (1-decene) (13), di-isooctyl phthalate (14), bis (2-ethyl hexyl) phthalate (15), dodecane (16), eicosane (17), heneicosane (18), 1,2-benzenedi carboxylic acid (19), lauric acid (20), and dodecanoic acid (21). According to the previous literature, these compounds were isolated from other aforementioned studies. 

Table 2 and Table 3 represents the name of the compound, molecular formula, molecular weight, area, quality, and activity. In addition, Figure 4 illustrates the compounds and their chemical structures. 

### 2.7. Gene Ontology and Pathway Analysis

Gene ontology analyses of hits with DAVID for the DNA microarray experiments using CAP and *cyclo*-(L-Val-L-Pro), as a representative for DKP revealed 485 genes upregulated and nine genes that were downregulated. A total of 74 genes were considered to be upregulated in the biological function group. Most categories were under the adenosine monophosphate (AMP) salvage pathways (25 gens), translation category (24 genes), metabolic process (13 genes), followed by an oxidation-reduction process (12 genes). Besides, nine genes were downregulated in the biological function under categories of the cell cycle, cell division, and cellular amino acid metabolic process. In addition, 153 genes were considered upregulated in the cellular function. The furthermost categories were under ribonucleoprotein complex (16 genes), ribosome (25 genes), and cytoplasm (19 genes). Moreover, 258 genes upregulated in the molecular function level.

Scattered under RNA binding (28 genes), transferase activity (20 genes), and structural constituent of ribosome (19 genes), oxidoreductase activity (15 genes), and catalytic activity (14 genes). The other genes, which were not documented, were under hypothetical proteins. The mode of action of PhACs against MRSA ATCC 43300 was an inhibition of translation process, as shown in (Figure 5).

### 2.8. Techno-Economic Analysis of the PhACs Productivity from SUK 25

The annual report functioning period for production PhACs in the SmF process is 3300 h/year (interchangeable for 330 handling days). The complete capital expenditure (TCI) for a recommended plant together with the secure capital guesstimate (FCE) and functioning working capital cost (WCC) (Equation (4)).
(4)TCI=FCE+WCC 

The FCE involves the cost of purchasing equipment, installation of the system, process piping, electronic systems, percussion and sensors, yard upgrades, buildings, and perhaps even the cost of WCC, which, as indicated by Herrera-Rodriguez et al. [34], may constitute 6.5% of the FCE. Consequently, the FCE to plan a production process with 1000 m^3^/day of aptitude reaches U.S. Dollar (U.S.D) 507,000.00, as described in Table 4.

The surplus treatment practice costs may reach 50% of the equipment cost, which is equivalent to U.S.D 84,500.00. Therefore, the TCI of the production unit could outstretch U.S.D 591,500.00. WCC is considered to be 6.5% of the FCE (U.S.D 38,447.50). As a result, permitting Equation (4), the TCI of U.S.D 629,947.50 could be premeditated. The TEA is considered one of the superlative methods to assess the efficacy of any eccentric scheme for PhACs productivity.

### 2.9. Annual Operation Cost

The annual operating cost (AOC) for the production, purification, and applications of PhACs consists of the cost of underdone materials (C-_RM_), production process waste (C-_WG_), utilities (C_-U_), and extra costs (C_-E_) such as extra maintenance or any other emergency things such as downing tools due to any technical defect in the equipment during the production of bioactive compounds as assessed on an anniversary basis in accordance with Equation (5).
(5)$$AOC=C_{RM}+C_{WG}+C_{U}+C_{E}$$

The $C_{RM}$ calculates the portion of raw materials used as a production medium and the chemicals needed for the production and purification manufactured by a domestic supplier. The$\;C_{U}$ comprises electricity and water that are mandatory for the operation progression and predictable grounded on the price for each component in the native exchange. The $C_{WG}$ characterizes the ultimate biomass yield produced in the production process of the PhACs. The AOC is for the production, purification, and applications of pharmaceutical active compounds. The raw supplies cost ($C_{RM}$) was probable at U.S.D 438,700.00/year, as described in Table 5. The estimated cost of the entire charge of utilities was U.S.D 100,000/year, as conveyed in China. This approximation is on the contrary reported in Malaysia. The operating labor entails 10 hands with a mediocre salary of U.S.D 120,000 year. The preservation and insurance were projected as 2 and 1% of the FCE, respectively, as reported Gunukula et al. [35].

### 2.10. PhACs Profitability and Annual Revenue

The present rate of seven PhACs that were produced and purified in the current work are *cyclo*-(tryptophanyl-prolyl) (U.S.D 40/100 mg), chloramphenicol (U.S.D 6/1000 mg), *cyclo*-(L-Val-L-Pro) (U.S.D 200/100 mg), *cyclo*-(L-Leu-L-Pro) (U.S.D 220/100 mg), *cyclo*-(L-Phe-L-Pro) (U.S.D 240/100 mg), *cyclo*-(L-Val-L-Phe) (U.S.D 250/100 mg), N-(7-hydroxy-6-methyl-octyl)-acetamide (U.S.D 200/100 mg).

The specific cost of these products $C_{Z}$ (kg/U.S.D) is the deliberated, which is dependent on the utility cost $C_{U}$, annual costs of capital $C_{C}$, raw material cost $C_{RM}$, extra cost $C_{E}$, and the annual PhACs production, as presented by [36] (Equation (6)).
(6)$$C_{Z}=(C_{C}+C_{U}+C_{RM}+C_{E})/E_{P}$$

The techno-economic analysis of the PhACs productivity from SUK 25 has revealed that at the optimum productivity of PhACs with 0.1 m^3^ was 17.142 g of *cyclo*-(tryptophanyl-prolyl), 14.27 g of chloramphenicol, 5 g of *cyclo*-(L-Val-L-Pro), 3.57 g of cyclo-(L-Leu-L-Pro), 2.857 g of *cyclo*-(L-Phe-L-Pro), 2.857 g of *cyclo*-(L-Val-L-Phe), and 4.285 g N-(7-hydroxy-6-methyl-octyl)-acetamide was generated. Thus, the 27.75 runs throughout the year in SmF with 100 m^3^ for each run, the total quantity generated consists of 475 kg of *cyclo*-(tryptophanyl-prolyl), 396 kg of chloramphenicol, 138.75 kg of *cyclo*-(L-Val-L-Pro), 99.1 kg of *cyclo*-(L-Leu-L-Pro), 79.2 kg of *cyclo*-(L-Phe-L-Pro), 79.2 kg of *cyclo*-(L-Val-L-Phe), and 118.9 kg N-(7-hydroxy-6-methyl-octyl)-acetamide.

The total income for manufacturing these quantities offers U.S.D 1,313,776.00. An approximation of the local tax in Malaysia is 15% (U.S.D 197,066.40) and AOC with FCE (U.S.D 673,910.00). The annual turnover (after the tax) of PhACs was U.S.D 442,799.60/year, as shown in Table 5. The average precise cost of 100 mg of PhACs was evaluated to be U.S.D 48.61 per 100 mg, which is low-priced compared with the market price of U.S.D 164.3/100 mg for the average. In this investigation, the 10-year records of the internal rate of return (IRR), payback period (PBP), and net present value (NPV) were gleaned for the assessment of the economics of PhACs production. The IRR designated the competence of the investment was more than 45% in this investigation, as shown in (Figure 6), which specifies that green PhACs are economically practicable. In addition, the cost of waste production will be calculated when this techno-economic analysis is applied for a future study.

## 3. Discussion

Different stages of optimization of fermentation have been analyzed using RSM. The key finding in this study was the statistical optimization of the significant media components that enhanced antibacterial activities against MRSA ATCC 43300 by using RSM. The model suggested that the glucose, asparagine, and mannitol affected the metabolite production from secondary metabolites produced by SUK 25, as reported with several earlier studies [37,38]. If the steepest descent experiments could be designed according to the findings of the dual-level factorial experiments, it will facilitate more insight into the optimal area of each significant factor. Moreover, the minor changes in media components, such as sources of carbon and nitrogen, or physical factors including agitation, aeration, temperature, fermentation incubation periods, and pH can considerably impact the quantity of the bacterial growth curve as in the level of log_10_ Colony-Forming Unit (C.F.U)/mL [39].

In this study, we determined the value of the secondary metabolites and metabolic profiles of associated microorganisms, their activity based on the size of inhibition zone diameter, and the values of MIC. There is a need to develop viable options, which involve the application of statistical methods for optimization with the capacity to correct existing problems related to conventional optimization [23]. The fact that any component required for the growth of microorganisms is a potential substrate adds to the complexity in culture media optimization [40]. Therefore, the improvement of antibiotic yield will entail a design of the correct medium and determining the necessary conditions for cultivation [41]. In a previous study, the one-at-a-time strategy design was applied to assess the impact of carbon and nitrogen sources, pH, and culture temperature on the production of bioactive compounds produced by SUK 25 strain as reported by Ahmad et al. [39]. On the contrary, the current study optimized the bioprocess using a combination of two methods. In the first method, the best nitrogen and carbon sources were selected for the growth of SUK 25 and precise measurement of the maximum weight of the crude extract, MIC, and maximum inhibition zone using the one-at-a-time strategy design. The findings from the present study are consistent with previous reports. The MIC values showed an improvement in the MIC values. Comparison with previous studies using *Nocardia* and *Streptosporangium* against *S. aureus* and MRSA showed that MIC against *S. aureus* were 30 μg and MRSA 40 μg [29]. The inhibition zone results are similar to a previous study carried by Mangzira Kemung et al. [42] where MUSC 125 towards MRSA ATCC 43300 showed the I.Z of 19 ± 0 mm, while the I.Z was 19.33 ± 0.58 mm against MRSA ATCC 33591.

The representations of the bonds of N–H and O–H stretching functional group represents carboxylic acids [43], and 2516, 2070 cm^−1^ bond H–C=O: C–H stretch represents the aldehydes functional group and were also attributed to C=C stretching of alkynes [44]. The peaks at 1558 cm−1 show the bond C–C stretch (in–ring) as the functional group for aromatics [45]. The peaks at 1449, 1394 cm^−1^ represent C–H bond, C–H rock as the functional group for alkanes. The peaks at 1121, 1092, and 972 cm^−1^ are due to the presence of C–O stretch (primary alcohol) and =C–H bend alkenes, O–H bending as carboxylic acids, and revealed the presence of C=O stretching of acid anhydrides [27]. The band at 821 cm^−1^ shows aromatic compounds [16].

The GC-MS analysis revealed that among the 38 peaks, 21 compounds that demonstrate the presence of bioactive constituents have been previously reported for their antimicrobial [33], antioxidant [14], antifungal [15], anti-inflammatory activates [18], and neurotropic action [18,19].

According to the gene ontology analyses of the biological function group, the most categories were under the adenosine monophosphate (AMP) salvage pathways, translation category, and metabolic process, followed by an oxidation-reduction process. Our findings are consistent with previous reports by Cui et al. [46]. The downregulated genes were under categories of the cell cycle, cell division, and cellular amino acid metabolic process, which corroborates the results from an earlier study by Guihua et al. [47]. The cellular function groups with the furthermost categories were under ribosome, cytoplasm, and ribonucleoprotein complex. These findings were published with the previous study that was conducted by Desaulniers et al. [48].

The most upregulated genes for the molecular function level were under categories of RNA binding, transferase activity, structural constituent of ribosome, oxidoreductase activity, and catalytic activity. These results are similar to the previous research conducted by Alam et al. [49].

The techno-economic analysis of the PhACs productivity from SUK 25 has revealed that the optimum productivity of PhACs with 0.1 m^3^ was cheaper than that in the market price. The average of the specific cost of 100 mg of PhACs was evaluated to be U.S.D 48.61 per 100 mg, the market price was U.S.D 164.3/100 mg for the average. These results are similar to the previous researches conducted by [35,36].

In this study, the techno-economic viability and limitations that need more attention, while seeing the development-associated technologies with the major steps of the production of PhACs such as cultivation, harvesting, collection, logistics, pretreatment, fermentation, and separation will be improved during the production of the PhACs at the industrial level. In conclusion, the PhACs production activity against MRSA ATCC 43300 was improved by 42.9% of crude extract, from 16 to 4 and 8 µg/mL for MIC, while the I.Z increased by 5.8 mm. The mechanism of CAP and DKP against MRSA acted by inhibiting protein synthesis, via preventing protein chain elongation by preventing the peptidyl transferase activity of the bacterial ribosome. The average specific cost of 100 mg of PhACs was evaluated to be cheaper than that at market price.

This study represents a preliminary study in the TEA, which is an initial exploration of the TEA technology by using SuperPro Designer software. The complete information will be in the next future study, which will calculate the amount of hot water, steam, heat, and waste production. In addition, it will calculate the flowrates of individual substances.

## 4. Materials and Methods

### 4.1. Strain and Culture Condition

The antibacterial compound producing strain, *Streptomyces* SUK 25, was previously isolated from the root of the *Zingiber spectabile* plant and extensively studied for their antibacterial secondary metabolites. The SUK 25 strain was maintained over the surface of ISP2 agar for 14 days before used [3].

### 4.2. Basal Production Medium

The SUK 25 was cultured on ISP2 agar media, followed by incubation at 28°C for 12 days. Then, a few blocks of ISP2 media containing pure and good growth of SUK 25 were transferred into 250 mL of modified Thornton’s medium, which is composed of (g/L) K_2_HPO_4_ 1.0, KNO_3_ 0.5, MgSO_4_·2H_2_0 0.2, CaCl_2_·H_2_O 0.1, NaCl 0.1, FeCl_3_ 0.01, asparagine 0.5, and glucose 1.5 and pH 7.4 [50].

### 4.3. RSM-Based Optimization of PHACs Production

The experimental setup in the present research included optimizing the internal and external factors of the submerged fermentation (SmF) with endophytic *Streptomyces* SUK 25, production, purification, and application of PhACs against MRSA ATCC 43300. The best operating parameters for PhACs production were assessed by response surface methodology (RSM) via central composite design (CCD), designed using the Design-Expert version 11 software. Factorial complete randomized design (CRD) (7×3×3) in triplicate was employed to examine the independent variables affecting the PhACs production. Seven independent factors including incubation time ($x_{1}$) (3–21 days), pH ($x_{2})\;$(4–9), ($x_{3}$) temperature (26–32 °C),$\;\left(x_{4}\right)\;$shaker rotation speed (120–180 rpm), ($x_{5}$) glucose concentration (1–3 g/L), ($x_{6}$) mannitol concentration (1–3 g/L), ($x_{7}$) asparagine concentration (0.5–2.5 g/L), and three points for each factor as well as three dependent variables were studied, and it included $y_{1}$(crude extracts), $y_{2}\;$minimum inhibitory concentration (MIC), and $y_{3}\;$inhibition zone (I.Z), as described in Table 6. The selection of the range of each independent factor was as described by Kang et al. [13]. Three-stage testing, high (+), medium, and low (−), was conducted for each independent factor as presented in Table 6.

A total of thirty-seven experimental runs were carried out simultaneously to study the efficiency of the selected factors on the production. For the outcomes to be optimized in terms of the independent variables, the quadratic model was implemented as the following Equation (7) reported by Noman et al. [51].
(7)$$y=\beta 0+\sum_{i=1}^{k}\beta_{i}x_{i}+\sum_{i=1}^{k}\beta_{ii}x_{i}{}^{2}+\overset{n}{\underset{i=1}{\sum }}\overset{n}{\underset{i<j}{\sum }}\beta_{ij}x_{i}x_{j}$$
where y represents the expected responses for the production process; coefficients of the linear model are represented as *β*_0_, *βi*, *βii*, and *βij*; coded independent factors = *x_i_*; while *k* represents the results of the responses.

The affiliation between the coded variable and experimental data is denoted as the following Equation (8):(8)$$x_{i}={ɛ}_{i}\;\frac{\left[\mathrm{HL}+\mathrm{LL}\right]/2}{\left[\mathrm{HL}-\mathrm{LL}\right]/2}$$
where *x_i_* denotes the coded variable, ${ɛ}_{i}$ is the experiment results, LL and HL denote the lowest and the highest level of the independent factors.

### 4.4. Submerged Fermentation and Extraction of Secondary Metabolites Methods

The submerged fermentation (SmF) process was carried out in a 250 mL Erlenmeyer flask containing Thornton’s medium (g/L), according to the composition presented by Alshaibani et al., 2016 [6]. The SmF pH medium was set to pH 7.4 using 0.1 M each of HCl and NaOH. The flasks were plugged with foam caps and enclosed with aluminum foil, before being autoclaved at 121 °C for 15 min. The autoclaved flasks were left to cool, then 100 microliter (μL) of SUK 25 spore suspension were inoculated, as described by Thakur et al. [49]. The 37 runs of the fermentation process were carried out according to the RSM design, as presented in Appendix A.

A Büchner funnel incorporated with Whatman^®^ Grade 1 filter paper was used to filter the production medium by using the vacuum filtration. Thereafter, 100 mL of the supernatant was mixed with half volume (50 mL) of ethyl acetate (EtOAC) in 3 separated times, and homogenized at 125 rotation/minutes (rpm) for 1 min. The separation and extraction of the PhACs from the supernatant was completed using 1 L separating funnels. The mixture of PhACs with EtOAC was separated by evaporation using a rotary evaporator RV 10 at 40 °C for 30 min and vacuum at 240 mbar [3].

All 37 runs were analyzed in triplicates, and the response was computed based on the mean antibacterial activity against MRSA ATCC 43300.

### 4.5. Determination of Dependent Variables and Determination of Antibacterial Activity

Three dependent variables were selected to be studied in this research according to the previous study reported by [25]. The first variable was $\left(y_{1}\right)$, which represents the weight of the crude extracts in milligram for each run. The weight of the crude extract obtained as ethanolic extract was determined using high-precision milligram balance (Intelligent-Lab™ Milligram Balance, PM series with 0.001 g accuracy); the means of the three runs were calculated. The second variable was$\;\left(y_{2}\right)$, which represents the MIC assay in µg/mL, and a 96-well plate was used for the MIC, which allowed a wider range of sample concentration and lower volume of samples to be tested. The third variable was $\left(y_{3}\right),\;$which represents that the inhibition zone in mm was performed against MRSA ATCC 43300, according to the Clinical and Laboratory Standards Institute (CLSI) M 07- A10 (2016) [52].

### 4.6. Purification of PhACs by Column Chromatography and Thin-Layer Chromatography

To purify the PhACs, the production process was conducted using SmF in a 1 L Erlenmeyer flask containing 800 mL Thornton’s medium based on the working parameters presented in Section 2.1. The procedure was performed at incubation time (day 12), pH 7, temperature 28 °C, shaker rotation speed (140 rpm), glucose (2 g/L), mannitol (1 g/L), asparagine (2 g/L), as recommended by RSM. The crude containing the PhACs were separated, as described in Section 2.1. Thereafter, air-dried powder samples of ethyl acetate crude extracts were processed by weighing and mixing them with silica powder (2 g) and dried at room temperature. This mixture was subjected to separation through column chromatography (CC) size exclusion of 2 × 40 cm containing 150 mg of the Sephadex LH_20_. A linear gradient of 100% chloroform was used for the starting gradient elution, and the polarity of methanol was gradually increased (100:0, 95:5, 90:10, 85:15, 80:20, 70:30, 60:40, 50:50, and 0:100 v: v) to 100% in 30 min. All the separates were dried upon column elution under lower pressure before using thin-layer chromatography (TLC) for another round of purification. Aluminum plates pre-coated with silica gel plates 60 F_254_ (20 × 20 cm, 0.25 thickness, MERCK) were used. The TLC plates were developed using a solvent system and a mobile phase consisting of ethyl acetate, hexane, and methanol (4:4:2 volume/volume/volume (v/v/v)). To visualize the chromatograms, a short ultraviolet (UV) (λ = 254 nm) and long-wavelength UV (λ365 nm) were used for the absorbance and fluorescence, respectively. Then, the chromatograms were sprayed with 10% H_2_SO_4_ in ethanol and heated at 100 °C for 10 min. The compounds with comparable R_f_ in TLC were mixed homogeneously, as reported by [53].

### 4.7. Fourier Transform Infrared Spectroscopy Spectrometer

The Fourier transform infrared spectroscopy (FT-IR) spectra was collected using a Perkin-Elmer-2000 spectrophotometer under a resolution of 4 cm^−1^ at 64 scans. A fixed weight (2 mg) of the crude extract and pure compounds were deposited in the KBr disks to create a thin film. The records of all spectra were conducted from the range 4000 to 400 cm^−1^ and analyzed with the computer software program, Spectrum for Windows (Perkin-Elmer), as reported by [54]. For every sample, triplicate spectra were processed and analyzed.

### 4.8. GC-MS Analysis of Volatile Components of Streptomyces SUK 25

The Shimadzu GC-MS QF2010 EI/NCI system was used to analyze the incomplete purified active fractions from SUK 25. Other parts incorporated with the system include a film thickness of 0.25 μm, ZB-5MS capillary column I.D. 30 m × 0.25 mm, and 5%-phenyl-arylene stationary phase, as previously reported by Sharma et al. [55]. Thereafter, 1.0 microliter (μL) of crude extract sample injected into the GC column for analysis, while maintaining the column and injector temperature at 70 and 200 °C, respectively. A split mode ratio of 40 and a flow rate of 1.51 mL/min were used. A similar flow rate (pressure of 105 kPa) was maintained for the carrier gas containing helium. The MS with ion 200 °C, interface temp: 240 °C, scan range: 40–1000 *m/z*, event time 0.5 sec, solvent cut time: 5 min, MS start time: 5 min, MS end time: 35 min. In addition, the ionization voltage was EI (-70 ev). A comparison between the mass spectra and the data from the National Institute of Standards and Technology, US (NIST05) was employed to identify the chemical compounds in the extract [56]. The software used in observing the chromatogram was again implemented for the final analysis, by determining and confirming the molecular weight, name, and peak area percentage of unknown compounds.

### 4.9. Gene Ontology and Pathway Analysis

The gene ontology, classified genes, gene products, and pathway analysis were conducted by using the web-based Kyoto Encyclopedia of Genes and Genomes (KEGG) databases and using the Database software for Annotation, Visualization, and Integrated Discovery (DAVID) (https://david.ncifcrf.gov, Accessed on: 13 November 2020).

### 4.10. Techno-Economic Analysis of the PhACs Productivity from SUK 25

#### 4.10.1. Process Simulation

The TEA was conducted for a facility that handles 1000 m^3^/day of dry crude extract production in SmF process, isolated, and purified from SUK 25.

The facility was assumed operating 3300 h/year (interchangeable for 330 handling days). The TEA was developed based on experimental data from the previous studies, as indicated by [57]. The methods proposed have three stages intended as the production (A), purification (B), and applications (C) as illustrated in (Figure 7).

#### 4.10.2. Production and Extraction of PhACs in SmF Media

In phase A, the production of PhACs in SmF process was conducted. To improve the PhACs productivity, the SmF process was performed under the proper operating parameters as presented previously in the Section 4.4.

#### 4.10.3. Column Chromatography and Thin-Layer Chromatography

In stage B, column chromatography and TLC were applied for the purification of PhACs, as presented in Section 4.6.

#### 4.10.4. Effectiveness of the PhACs and Determination of Antibacterial Activity

In stage C, the effectiveness of the PhACs against the MRSA ATCC 43300 is carried out, as presented in Section 4.5.

#### 4.10.5. Economic Analysis

The models built in the techno-economic analysis was applied by using the SuperPro Designer for the production of seven pure active compounds isolated and identified in our previous research from SUK 25 against MRSA ATCC 43300, as represents the first attempt of a TEA of production of PhACs from SUK 25. The proposed flowchart of PhACs was then established to estimate the capital and operating costs of each process. Table 4 details the fixed capital estimate (FCE) for production of PhACs by the SUK 25 using SmF and the other economic assumptions made in the study.

The cell culture bio-reactor, centrifuge, purification system, freeze drying, inoculum preparation, storage tank, microfiltration, aerobic bio-oxidation, dead end filtration, and biomass storage tank. In addition, the equipment installation, process piping, instrumentation and controls, electrical systems, buildings, yard improvements, and construction were considered in the analysis. This approach assumes that the technology is mature and that several of facilities employing the same process are established and are fully operational. Vendor quotes were used when the costs were not available for the row material [58]. The total direct costs were estimated as the sum of all the installed equipment costs, plus the costs for buildings, piping, and site development, as shown in Table 4.

#### 4.10.6. Annual Operation Cost

The annual operating cost (AOC) for the production, purification, and applications of PhACs consists of the cost of underdone materials (C-_RM_), production process waste (C-_WG_), utilities (C_-U_), and extra costs (C_-E_). The operating costs that include variable costs (feedstocks, utilities, etc.), fixed costs (salaries, maintenance, taxes, etc.), and general costs (e.g., marketing expenses) were estimated based on the principles, as reported by [58]. The C-_U_ comprises electricity and water that are mandatory for the operation progression and predictable grounded on the price for each component in the native exchange. The C__WG_ characterizes the ultimate biomass yield produced in the production process of the PhACs. Techno-economic problems derive into demonstration when a procedure is scaled-up or needs to be scaled-up to a marketable level. The main problem to be kept in mind is choosing the right optimized parameters, which are measured for cost economics. Reagents required for the purification and reagents required for the application test, electricity, and water as well as equipment used in the bioprocess are inclusive in the calculation. The sensitivity analysis was laboring to evaluate the influence that different conditions may have on the price of the purified PhACs.

### 4.11. Statistical Analysis

The linear effect of the independent factors and interactions between these factors were examined using analysis of variance (ANOVA), and *p*-values ≤ 0.05 were considered significant. The efficiency of SmF for producing PhACs was investigated as a function for the quadratic model and coefficient adjusted R (R^2^ adj.). All experiments were conducted in triplicate. Three-dimensional (3D) graphical illustrations were used to depict the association between factors and the degradation process. To show the association between the experimental phases of each explanatory variable, 3D response surface plots were produced.

## 5. Conclusions

The most recognizable finding to appear from this study is that the PhACs’ production activity against MRSA ATCC 43300 was improved by 42.9% of dry weight of the crude extract; MIC also enhanced from 16 to 8 µg/mL, while the I.Z increased by 5.8 mm. The GC-MS analysis revealed that 21 compounds that demonstrate the presence of bioactive constituents were identified. The mechanism of CAP and DKP against MRSA acted by inhibiting protein synthesis, via preventing protein chain elongation by preventing the peptidyl transferase activity of the bacterial ribosome. The average specific cost of 100 mg of PhACs was evaluated to be cheaper than that at the market price.

## Figures and Tables

**Figure 1 molecules-26-02510-f001:**
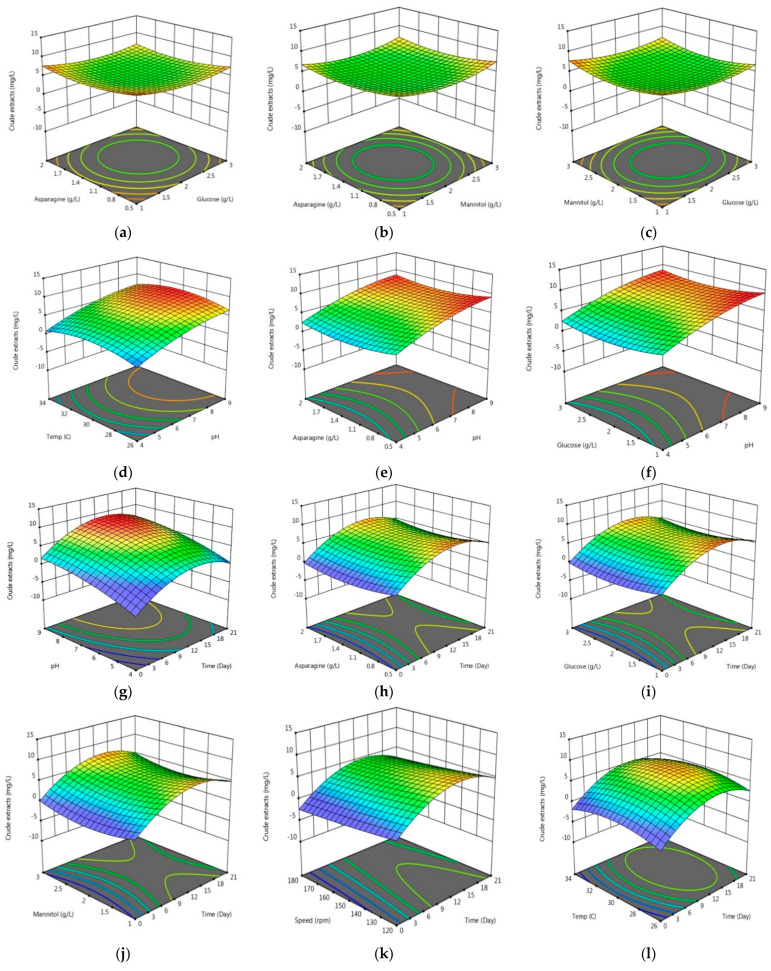
Three-dimensional response surface plot for interactions between $x_{1}$ (time) (day), $x_{2}$ (pH), $x_{3}$ (temperature) (°C), $x_{4}$ (speed) (rpm), $x_{5}$ (glucose) (g/L), $x_{6}$ (mannitol) (g/L), $x_{7}$ (asparagine) (g/L), and their effects on $y_{1}$ (crude extracts) (mg/L). (**a**) Effects of asparagine and glucose on weight of crude extract; (**b**) Effects of asparagine and mannitol on weight of crude extract; (**c**) Effects of mannitol and glucose on weight of crude extract; (**d**) Effects of temperature and pH on the weight of crude extract; (**e**) Effects of asparagine and pH on weight of crude extract; (**f**) Effects of glucose and pH on the weight of crude extract; (**g**) Effects of pH and time on the weight of crude extract; (**h**) Effects of asparagine and time on the weight of crude extract; (**i**) Effects of glucose and time on the weight of crude extract; (**j**) Effects of mannitol and time on the weight of crude extract; (**k**) Effects of speed and time on the weight of crude extract; (**l**) Effects of temperature and time on the weight of crude extract.

**Figure 2 molecules-26-02510-f002:**
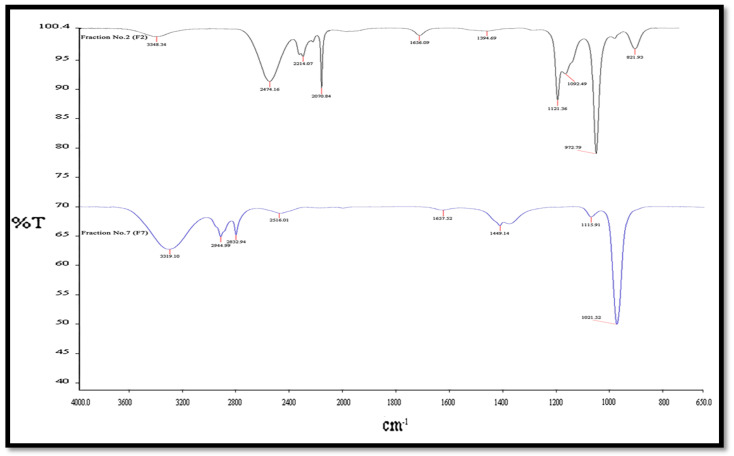
The Fourier transform infrared spectroscopy for fraction number 2 and 7.

**Figure 3 molecules-26-02510-f003:**
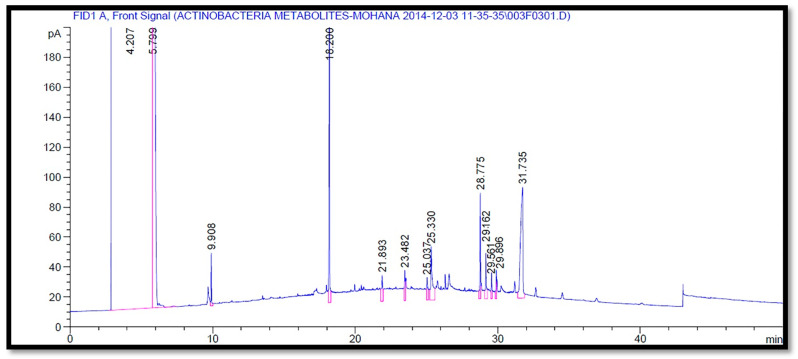
The GC-MS chromatograms analysis of volatile organic compounds of the ethanolic crude extract in fraction number 2 at different retention times.

**Figure 4 molecules-26-02510-f004:**
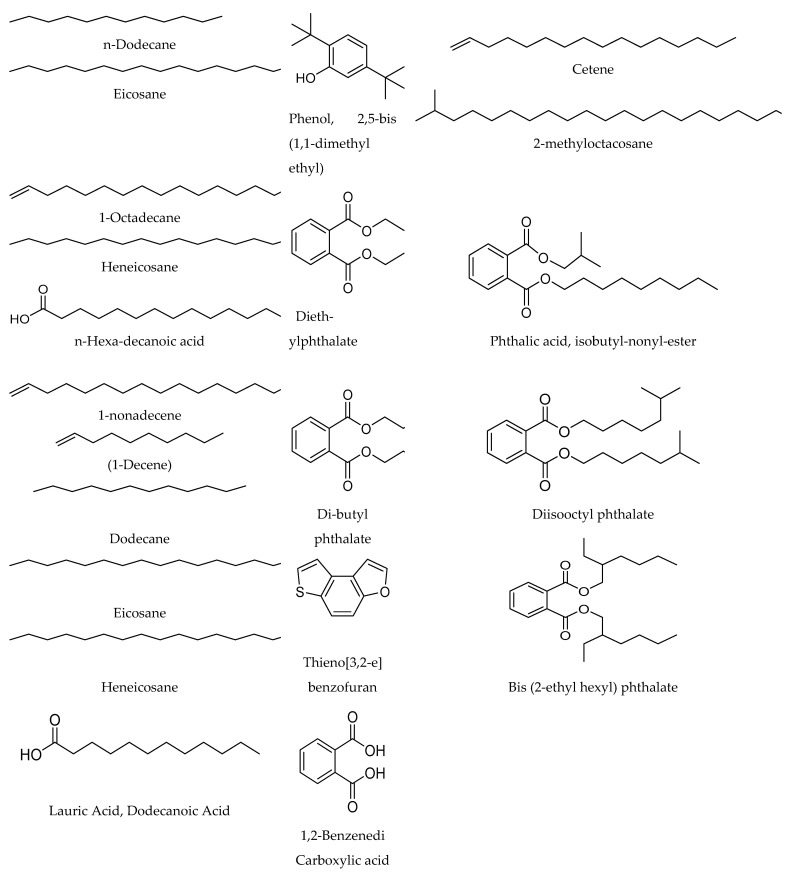
The chemical structure of the antimicrobial compounds identified by GC-MS from crude extract of SUK 25.

**Figure 5 molecules-26-02510-f005:**
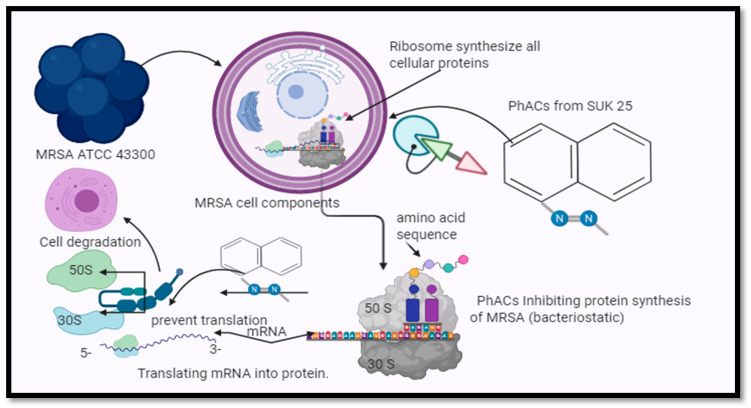
The mode of action of PhACs from SUK 25 against MRSA ATCC 43300. PhACs from SUK 25 stop the translation process and prevent the cell to converts genetic information carried in an mRNA molecule.

**Figure 6 molecules-26-02510-f006:**
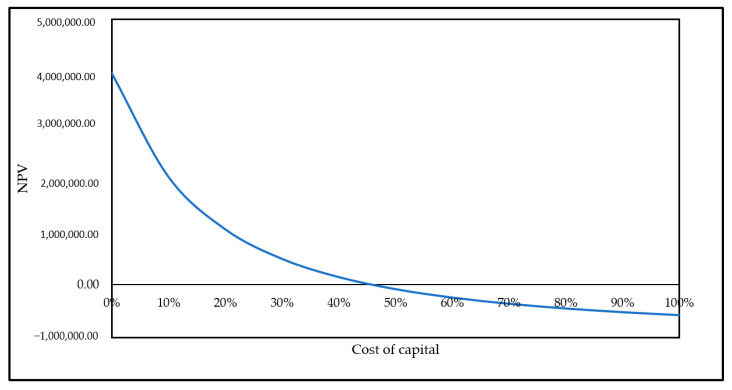
The IRR of the investment was more than 50% in this investigation.

**Figure 7 molecules-26-02510-f007:**
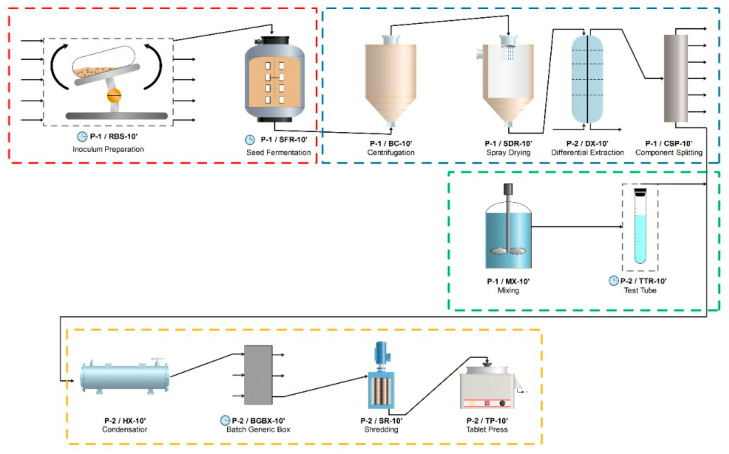
The proposed flowchart of PhACs production in SmF process.

**Table 1 molecules-26-02510-t001:** Analysis of the variance (ANOVA) of the response surface quadratic model for pharmaceutically active compounds production from *Streptomyces* SUK 25.

Source	DF	Sum of Squares			Mean Square			F Value			*p*-Value		
		$y_{1}$	$y_{2}$	$y_{3}$	$y_{1}$	$y_{2}$	$y_{3}$	$y_{1}$	$y_{2}$	$y_{3}$	$y_{1}$	$y_{2}$	$y_{3}$
Model	14	6.98	636.49	23.16	6.98	636.49	23.16	3.80	2.70	4.83	0.0026 Significant	0.0180 Significant	0.0005 Significant
Residual error	22	1.84	235.51	4.80	1.84	235.51	4.80						
Lack-of-fit	16	2.40	211.33	5.82	2.40	211.33	5.82	7.20	0.7044	2.79	0.0112 Significant	0.7327 Not significant	0.1053 Not significant
Pure error	6	0.3333	300.00	2.08	0.3333	300.00	2.08						
Total	36												

$x_{1}$ (time) (day),$\;x_{2}$ (pH), $x_{3}$ (temperature) (°C),$\;x_{4}$ (speed) (rpm), $x_{5}$ (glucose) (g/L), $x_{6}$ (mannitol) (g/L), $x_{7}$ (asparagine) (g/L), $y_{1}$ (crude extracts) (mg/L), $y_{2}($(MIC) (µg/mL),$\;y_{3}($(I.Z) (mm). $y_{1}$ R^2^=0.70; R^2^ (adj.) = 0.52,$y_{2}$ R^2^ = 0.63; R^2^ (adj.) = 0.40, $y_{3}$ R^2^ = 0.75; R^2^ (adj.) = 0.6.

**Table 2 molecules-26-02510-t002:** Antimicrobial compounds identified in the ethyl acetate extract by GC-MS from fraction no. 2.

Peak no	R. Time	Name of the Compound	Molecular Formula	Molecular Weight (g/mol)	Area %	Quality (%)	Activity	References
1	5.799	n-dodecane	C_12_H_26_	170	0.66	76	Antioxidants, antimicrobial	[14,15]
2	9.908	Eicosane	C_20_H_42_	283	1.38	74	Antibacterial, antifungal	[16]
3	10.200	Phenol, 2,5-bis (1,1-dimethyl ethyl)	C_14_H_22_O	206	2.24	87	Antimicrobial	[17,18]
4	21.893	Cetene	C_16_H_32_	29	0.82	75	Antioxidants	[19]
5	23.482	Diethylphthalate	C_12_H_14_O4	222	3.92	88	Antimicrobial	[16]
6	25.037	2-methyloctacosane	C_29_H_60_	409	0.81	94	Antifungal	[20]
7	25.330	1-octadecane	C_18_H_36_	252	0.86	75	Antifungal	[21]
8	28.775	Phthalic acid, isobutyl nonyl ester	C_21_H_32_O_4_	348	1.11	79	Antimicrobial, antioxidants	[22]
9	29.896	Heneicosane	C_21_H_44_	297	1.86	74	Antibacterial	[23]
10	31.735	n-hexa-decanoic acid	C_16_H_32_O_2_	256	1.02	73	Cosmetics, antioxidants	[24,25]

**Table 3 molecules-26-02510-t003:** Antimicrobial compounds identified in the ethyl acetate extract by GC-MS from fraction no. 7.

Peak no	R. Time	Name of the Compound	Molecular Formula	Molecular Weight (g/mol)	Area %	Quality (%)	Activity	References
1	17.671	Di-butyl phthalate	C_16_H_22_O_4_	278	2.64	86	Antifungal	[26]
2	12.301	1-nonadecene	C_19_H_38_	267	0.85	79	Antioxidants, antimicrobial	[27]
3	12.734	Thieno[3,2-e] benzofuran	C_10_H_6_OS	174	1.20	92	Antibacterial	[28]
4	13.279	(1-decene)	C_10_H_20_	140	0.46	85	Antifungal	[29]
5	13.509	Diisooctyl phthalate	C_24_H_38_O_4_	391	1.11	98	Anticancer, antibacterial	[30]
6	13.590	Bis (2-ethyl hexyl) phthalate	C_24_H_38_O_4_	391.56	79.06	94	Antimicrobial	[31]
7	14.084	Dodecane	C_12_H_26_	170	19.41	69	neurotropic action	[14,15]
8	17.762	Eicosane	C_20_H_42_	283	5.31	71	Antimicrobial, Antifungal	[16]
9	18.532	Heneicosane	C_21_H_44_	296	3.70	72	Antifungal	[23]
10	19.648	1,2-benzenedi carboxylic acid	C_8_H_6_O_4_	166.14	3.1	93	Antimicrobial and anti-inflammatory Activities	[32]
11	25.431	Lauric acid, dodecanoic acid	C_12_H_24_O_2_	200.32	2.1	74	Antimicrobial and anti-inflammatory Activities	[33]

**Table 4 molecules-26-02510-t004:** The fixed capital estimate (FCE) for production of PhACs by the SUK 25 using the submerged fermentation process.

Item Code *	Quantity	Item	Percentage of FCE	Cost
P-1/BR-101	1	Cell culture bio-reactor	33.33%	10,000.00
P-1/DS-101	2	Centrifuge		30,000.00
P-1/MSX-101	1	Purification system		17,000.00
P-1/FDR-101	1	Freeze drying		5000.00
P-1/RBS-101	1	Inoculum preparation		5000.00
P-2/V-101	2	Storage tank		25,000.00
P-1/MF-101	1	Microfiltration		14,000.00
P-1/AB-101	1	Aerobic bio-oxidation		30,000.00
P-2/DE-101	1	Dead end filtration		20,000.00
P-1/DB-101	1	Biomass storage tank		13,000.00
		Total equipment purchase cost		169,000.00
		Equipment installation	9.86%	50,000.00
		Process piping	9.86%	50,000.00
		Instrumentation and controls	9.47%	48,000.00
		Electrical systems	9.86%	50,000.00
		Buildings	7.89%	40,000.00
		Yard improvements	3.94%	20,000.00
		Construction	15.78%	80,000.00
		TOTAL		507,000.00

* Refer the item code to (Appendix A).

**Table 5 molecules-26-02510-t005:** Annual operation cost of the production of PhACs in submerged fermentation unit using Thornton’s medium after optimization via RSM.

	Component	Price (U.S.D)	Unit	Quantity	Total Cost (U.S.D)
**Raw material**	K_2_HPO_4_	80	kg	1000 kg	80,000.00
**(chemicals)**	KNO_3_	100	kg	500 kg	50,000.00
	MgSO_4_.2H_2_O	300	kg	200 kg	60,000.00
	CaCl_2_.H_2_O	100	kg	100 kg	10,000.00
	NaCl	100	kg	100 kg	10,000.00
	FeCl_3_	350	kg	10 kg	3500
	Glucose	415	kg	1500 kg	62,200.00
	Asparagine	1500	kg	50 kg	75,000.00
	HCl	100/L	1 L	100 L	10,000.00
	NaOH	160/L	1 L	50 L	8000
**Reagents required for the purification**					20,000.00
**Reagents required for the application test**					50,000.00
**Utilities**	Electricity	0.04	kWh	1,000,000	40,000.00
	Water	0.01	U.S.D/m^3^	6,000,000	60,000.00
**Other costs**	Labor	12000	U.S.D/	10	120,000.00
			employee		
	Maintenance	2	% of FCE		5070
	Insurance	1	% of FCE		10,140.00
**Total**					673,910.00

**Table 6 molecules-26-02510-t006:** The min and max range for each independent factor used for pharmaceutically active compounds production from *Streptomyces* SUK 25.

Medium Components	Codes	High Level (+)	Medium Level	Low Level (–)
Incubation Time (day)	$x_{1}$	21	12	0
pH	$x_{2}$	9	6	4
Temperature °C	$x_{3}$	26	30	34
Shaker rotation speed (rpm)	$x_{4}$	180	160	120
Glucose (g/L)	$x_{5}$	3	2	1
Mannitol (g/L)	$x_{6}$	3	2	1
Asparagine (g/L)	$x_{7}$	2	1.0	0.5

## Data Availability

The data presented in this study are available in the article and in the Appendix A online at www.mdpi.com/xxx/s1 and https://drive.google.com/file/d/1-DxbaSBpAia0q38SVc78LltY3WddPoNa/view?usp=sharing.

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
