# Peer review of "Enhanced Pharmaceutically Active Compounds Productivity from Streptomyces SUK 25: Optimization, Characterization, Mechanism and Techno-Economic Analysis"

_molecules, 2021, doi:10.3390/molecules26092510_

Round 1

Reviewer 1 Report

The manuscript show high level in originality and quality of presentation; great significant of content and interest to the readers including future perspectives...Compliments

Please, to verify the references list and to correct some text editing (i.e. capital letter on ref. 13, 17, 47, 52, 59) and Italic for microorganisms (i.e. ref. 4, 14, 18, 28...).

Author Response

Response to Reviewers

Reviewers comments to Article No. (molecules 1162661).

Thank you for giving us the opportunity to submit a revised draft of the manuscript “Enhanced pharmaceutically active compounds productivity from Streptomyces SUK 25; optimization; characterization, mechanism, and techno-economic analysis” for publication in the Journal of Molecules. We appreciate the time and effort that you and the reviewers dedicated to providing feedback on our manuscript and are grateful for the insightful comments on and valuable improvements to the manuscript. We have incorporated most of the suggestions made by the reviewers. Those changes are highlighted within the manuscript. Please see below, in red, for a point-by-point response to the reviewers’ comments and concerns. All page numbers refer to the revised manuscript file with tracked changes

Author's Reply to the Review Report (Reviewer 1)

Comments and Suggestions for Authors

  • The manuscript shows high level in originality and quality of presentation; great significant of content and interest to the readers including future perspectives...Compliments
  • Author response: Thank you!
  • Please, to verify the references list and to correct some text editing (i.e. capital letter on ref. 13, 17, 47, 52, 59) and Italic for microorganisms (i.e. ref. 4, 14, 18, 28...).
  • Author response: All references were corrected as you recommended. As shown in the manuscript.

Reference number

Line number

4

585

13

609

14

612

17

621

18

424

28

652

47

707

52

720

59

739

Reviewer 2 Report

The manuscript itself brings novelty to enhance active compound productivity from SUK25. I have these comments on the manuscript.

1) Eq.1-3 present mutual relationships among factors affecting crude extracts and MIC and IZ values. Is there any physical explanation, natural behaviour why such a linear or quadratic influence? It is food for thought concerning a more profound discussion to respond "why".

2) Regarding Eq.1-3, I recommend substituting x and y parameters directly by proper variables (time, pH, T, rpm,..). This improves the readability and clearness of presented dependences.

3) Can you suggest sensitivity analysis of Eq.1-3 and define which parameters are dominant to influence presented yields?

4) No comments on experimental set-up and results are listed in the manuscript. This is only mentioned in the supplementary material. I recommend adding the chapter that summarizes the essential information about the experimental set-up and results in the manuscript.

5) Can you discuss the error of experimental results? Did you try to perform several repetitions to reach identical yields?

6) Regarding the S3 table, the model's standard error is higher than the given coefficient. It seems that the model is very rough and does not fit appropriately to experimental data.

7) Techno-economic assessment is very poor because the equipment prices are presented without any technical set-up information. TEA must be defined by PFD scheme, mass and energy balances, fundamental size characteristics of installed equipment. Then how CAPEX and OPEX were found etc. TEA must be remarked to be fully defined and described.

8) The experimental set-up is placed on page 14. It does not make sense. Firstly present experiments and data, then present modelling approach and finally TEA. The structure of the manuscript must be rearranged.

9) I highly recommend grammar and stylistic language corrections.

Based on the information above, I recommend the major revision of the paper.

Author Response

Response to Reviewers

Reviewers comments to Article No. (molecules 1162661).

Thank you for giving us the opportunity to submit a revised draft of the manuscript “Enhanced pharmaceutically active compounds productivity from Streptomyces SUK 25; optimization; characterization, mechanism, and techno-economic analysis” for publication in the Journal of Molecules. We appreciate the time and effort that you and the reviewers dedicated to providing feedback on our manuscript and are grateful for the insightful comments on and valuable improvements to the manuscript. We have incorporated most of the suggestions made by the reviewers. Those changes are highlighted within the manuscript. Please see below, in red, for a point-by-point response to the reviewers’ comments and concerns. All page numbers refer to the revised manuscript file with tracked changes.

Author's Reply to the Review Report (Reviewer 2)

Comments and Suggestions for Authors

  • The manuscript itself brings novelty to enhance active compound productivity from SUK25.
  • Author response: Thank you!
  • I have these comments on the manuscript.

  • 1-3 present mutual relationships among factors affecting crude extracts and MIC and IZ values. Is there any physical explanation, natural behaviour why such a linear or quadratic influence? It is food for thought concerning a more profound discussion to respond "why".

Author response: It is known that the cultivation conditions greatly affect the secondary metabolites manufacture in Streptomyces. This study focus on the optimization of culture conditions factors to enhance the antibacterial efficacy of strain SUK 25 to produce more secondary metabolites with low MIC and good inhibition zone.  Different stages of optimization of fermentation media have been analyzed using RSM as statistical optimization of the significant media components that enhanced antibacterial activities against MRSA. The minor changes in the media components such as sources of carbon and nitrogen, or physical factors including agitation, aeration, temperature, fermentation incubation periods, and pH can considerably impact the quantity of the bacterial growth curve as in the level of log10 CFU/ml and amount of crude extract.  this study showed that there are relationships among factors in positive or negative way between the physical factors and the media components.  

2) Regarding Eq.1-3, I recommend substituting x and y parameters directly by proper variables (time, pH, T, rpm,). This improves the readability and clearness of presented dependences.

Author response:  we have tried to substitute the x with the direct parameters name but we faced a problem with the using x2 so we have listed the real names of the independent factors after the eqs.

3) Can you suggest sensitivity analysis of Eq.1-3 and define which parameters are dominant to influence presented yields?

Author response:  In the optimization medium of the growth of microorganism several factors such as internal and external factors are interfere and the most factors influence in the production of bioactive compounds are the carbon and nitrogen source and its concentrations.  In addition, the external factors such as pH, Temp., and rotation have influence and interfere with the production of bioactive compounds. We have highlighted (*) the independent factors which has significant role in optimization process.

4) No comments on experimental set-up and results are listed in the manuscript.

This is only mentioned in the supplementary material. I recommend adding the chapter that summarizes the essential information about the experimental set-up and results in the manuscript.

  • Author response: the experimental set-up was corrected as the reviewer recommended in the section Material and methods.  From line 409- 471.

5) Can you discuss the error of experimental results?

Author response: The standard error of regression was used to detect the fitting the experimental and predicted results. The value of standard error of regression recorded in this study ranged from 0.49 to 2.17 for (and from 0.79 to 3.52 for indicate the accuracy of the experimental results. (line 127- 133). Definitely, future investigation could focus on the more optimization of the culture conditions to enhance the Pharmaceutically bioactive compounds from SUK 25 needed for the industrial level.

Did you try to perform several repetitions to reach identical yields?

Author response: This experiments were carried out 37 runs of the fermentation process were carried out according to RSM design as presented in Table S1 in triplicate at each time to avoid any biological variation or technical errors.  In addition, the experiments were carried out during the optimization of growth curve of the SUK 25 for several times to determine the best environmental conditions and effect of physical factors.   In our optimization method we used previously The one-variable-at-a-time approach design, but this it  is time-consuming and tasking since the optimization method entails the use of a single parameter for every trial.

6) Regarding the S3 table, the model's standard error is higher than the given coefficient. It seems that the model is very rough and does not fit appropriately to experimental data.

Author response: In contrast, the standard error of  was between 8.22 and 24.63 this error is high and might be related to the  nature of the response and measurement methods.  In addition, this is because the low MIC is better than the high MIC (i.e.) when the MIC decrease it means the bioactive compound is better and more effective.  In this experiment we use the MIC as multiplex of (4). 

7) Techno-economic assessment is very poor because the equipment prices are presented without any technical set-up information. TEA must be defined by PFD scheme, mass and energy balances, fundamental size characteristics of installed equipment. Then how CAPEX and OPEX were found etc. TEA must be remarked to be fully defined and described.

Author response: In this study we have calculated all the chemical and operating cost manually since we have not full package of the software and the TEA does not show minor details such as piping details and designations. We used the SuperPro Designer which is used extensively in the pharmaceutical, biotech, specialty chemical, food, consumer product, metallurgical, and related industries.

(Added this in line 403): The comments are important for us to do the TEA by using the PFD scheme in our future researches.  In this study the techno-economic viability and limitations that need more attention while seeing the development associated technologies with the major steps of the production of PhACs such as cultivation, harvesting, collection, logistics, pretreatment, fermentation, separation will be improve during the production of the PhACs in the industrial level.  

8) The experimental set-up is placed on page 14. It does not make sense.

Firstly, present experiments and data, then present modelling approach and finally TEA. The structure of the manuscript must be rearranged.

Author response: experimental set-up was corrected as reviewer recommended from line (409 to 553). 

Material and methods

Line 410- 414

4.1 Strain and Culture Condition

Line 415 – 420

4.2 Basal Production Medium

Line 422-

4.3 RSM-Based Optimization of PHACs Production

9) I highly recommend grammar and stylistic language corrections.

- Author response: the manuscript sent for English proofreading and the official latter from the English center will attached.

- Based on the information above, I recommend the major revision of the paper.

  • Thank you for your comments and recommended.  Which are grateful for the insightful comments on and valuable improvements to the manuscript.
  •  

    The addition of new information according to reviewer comments.

    Line 116 – 119.

    The best and wide I.Z () was related positively and statistically significant (p <0.05) with factors, while having a non-significant correlation with factors .

    Line 127- 133

    The standard error of regression was used to detect the fitting the experimental and predicted results. The value of standard error of regression recorded in this study ranged from 0.49 to 2.17 for amount of crude extracts (and from 0.79 to 3.52 for I.Z which indicate the accuracy of the experimental results. In contrast, the standard error of  was between 8.22 and 24.63 this error was high and might be related to the nature of the response and measurement methods where the MIC is represent good response when it was in low value. 

    Line 143 – 146

    where; * represent the factors  that have a significant role;   (time) (day), (pH),  (temperature) (°C), (speed) (rpm),  (glucose) (g/L),  (mannitol) (g/L),  (asparagine) (g/L),  (crude extracts) (mg/L),  (MIC) (µg/mL), (IZ) (mm).

    line (290 – 292)

     such as for maintenance or any other Emergency things such as downing tools due to any technical defect in the equipment during production of bioactive compounds.

     Material and methods

    Line 410- 414

    4.1 Strain and Culture Condition

    Line 415 – 420

    4.2 Basal Production Medium

    Line 422-

    4.3 RSM-Based Optimization of PHACs Production

Reviewer 3 Report

This paper describes the enhancement pharmaceutically active compounds productivity from Streptomyces SUK 25. The aim of the study is important. Optimization, characterization and mechanisms were determined. Techno-economic analysis was estimated. Following revision should be considered before publication.

Minor revision

Figure 2, 3: Resolution of Figures should be improved.

Figure 4: Described with name of the chemical compounds.

Eq (5)  CE ) ?

Eq (6)  (CE  ?

Table 6 Write to how did you select of the range of each independent factor briefly.

Author Response

Response to Reviewers

Reviewers comments to Article No. (molecules 1162661).

Thank you for giving us the opportunity to submit a revised draft of the manuscript “Enhanced pharmaceutically active compounds productivity from Streptomyces SUK 25; optimization; characterization, mechanism, and techno-economic analysis” for publication in the Journal of Molecules. We appreciate the time and effort that you and the reviewers dedicated to providing feedback on our manuscript and are grateful for the insightful comments on and valuable improvements to the manuscript. We have incorporated most of the suggestions made by the reviewers. Those changes are highlighted within the manuscript. Please see below, in red, for a point-by-point response to the reviewers’ comments and concerns. All page numbers refer to the revised manuscript file with tracked changes. 

Author's Reply to the Review Report (Reviewer 3)

Comments and Suggestions for Authors

This paper describes the enhancement pharmaceutically active compounds productivity from Streptomyces SUK 25. The aim of the study is important. Optimization, characterization and mechanisms were determined. Techno-economic analysis was estimated. Following revision should be considered before publication.

  • Minor revision
  • Author response: Thank you!
  • Figure 2, 3: Resolution of Figures should be improved.

  • Author response: The figures were improved as shown in pages 8 line 238 to 242.

Figure 4: Described with name of the chemical compounds.

Author response: All compounds now with the name. The number were removed. Line 279.

(C-E) in Eq (5) and Eq (6)?

Author response:  extra costs (C-E)

Added in line (290 – 292) such as for maintenance or any other Emergency things such as downing tools due to any technical defect in the equipment during production of bioactive compounds.

Table 6 Write to how did you select of the range of each independent factor briefly.

Author response:  The values of three levels were set according to our preliminary experimental results using one at a time strategy. (Line 343- line 346) Therefore, the improvement of antibiotic yield will entail a design of the correct medium and determining the necessary conditions for cultivation [27]. In a previous study, the one-at-a-time strategy design was applied to assess the impact of carbon and nitrogen sources, pH, and culture temperature on the production of bioactive compounds by SUK 25 strain Ahmad et al., [25].

In addition, several previous studies selected the same rang of the independent factor.  The selection of the range of each independent factor was as described by Kang et al., [13].

  1. Kang, C., T.-C. Wen, J.-C. Kang, Z.-B. Meng, G.-R. Li, and K.D. Hyde, Optimization of large-scale culture conditions for the production of cordycepin with Cordyceps militaris by liquid static culture. The Scientific World Journal.2014. DOI: 10.1155/2014/510627.
  2. Ahmad, S.J., S. Suhaini, H.M. Sidek, D.F. Basri, and N.M. Zin, Anti-methicillin resistant Staphylococcus aureus activity and optimal culture condition of Streptomyces sp. SUK 25. Jundishapur Journal of Microbiology.2015, 8 (5).e16784. DOI: 10.5812/jjm.16784. doi: 10.5812/jjm.16784.

The addition of new information according to reviewer comments.

Line 116 – 119.

The best and wide I.Z () was related positively and statistically significant (p <0.05) with factors, while having a non-significant correlation with factors .

Line 127- 133

The standard error of regression was used to detect the fitting the experimental and predicted results. The value of standard error of regression recorded in this study ranged from 0.49 to 2.17 for amount of crude extracts (and from 0.79 to 3.52 for I.Z which indicate the accuracy of the experimental results. In contrast, the standard error of  was between 8.22 and 24.63 this error was high and might be related to the nature of the response and measurement methods where the MIC is represent good response when it was in low value. 

Line 143 – 146

where; * represent the factors  that have a significant role;   (time) (day), (pH),  (temperature) (°C), (speed) (rpm),  (glucose) (g/L),  (mannitol) (g/L),  (asparagine) (g/L),  (crude extracts) (mg/L),  (MIC) (µg/mL), (IZ) (mm).

line (290 – 292)

 such as for maintenance or any other Emergency things such as downing tools due to any technical defect in the equipment during production of bioactive compounds.

 Material and methods

Line 410- 414

4.1 Strain and Culture Condition

Line 415 – 420

4.2 Basal Production Medium

Line 422-

4.3 RSM-Based Optimization of PHACs Production

Round 2

Reviewer 2 Report

Dear authors. The manuscript was improved in the experimental part. Nevertheless, I have still engineering problems with TEA. Reading the manuscript, I am able to identify the production and incomplete TEA pieces. There is no PFD/block diagram of your solution in the manuscript. The flowrates of individual substances are not defined, mass/volumetric flowrates are missing. The CAPEX is always estimated knowing characteristic sizes of equipment (storage tank = volume, pump = flowrate, reactor = volume, filter = surface of filtration cloth etc.). But there is no information about it (the software should generate it). Just equipment prices are presented without any background. What about utilities, hot water, steam, heat. Just needed money is mentioned without any information on what and in which quantity is used. What about the waste production of your technology? There is still plenty of missing information to find your TEA reliable.

Author Response

Reviewers comments to Article No. (molecules 1162661).

Author’s Reply to the Review Report (Reviewer 2)

  • Reviewer Comment: Dear authors. The manuscript was improved in the experimental part. Nevertheless, I have still engineering problems with TEA. Reading the manuscript, I am able to identify the production and incomplete TEA pieces.

Authors' Responses: Thank you so much for valuable comments. The part of Techno-Economic Analysis was rearrangement as shown in section 4.10:  line 550.

4.10. Techno-Economic Analysis of the PhACs Productivity from SUK 25

  • Reviewer Comment: There is no PFD/block diagram of your solution in the manuscript.
  • Authors' Responses: The block diagram was in the supplementary data. We added it in the manuscript as Figure 7 Line 626 page 10.

  • Reviewer Comment: The flowrates of individual substances are not defined; mass/volumetric flowrates are missing. The CAPEX is always estimated knowing characteristic sizes of equipment (storage tank = volume, pump = flowrate, reactor = volume, filter = surface of filtration cloth etc.). But there is no information about it (the software should generate it).

Authors' Responses: Thank you for this comment. It is true however, as previously mentioned, we used the software that was not completely generate the all information of the capital expenditure. We reported only this statement.  (line 267. 276).

(The complete capital expenditure (TCI) for a recommended plant together with the secure capital guesstimate (FCE) and functioning working capital cost (WCC) (Eq. 4). 

TCI=FCE+WCC                                                                                                 (4)

The FCE involves the cost of buying equipment, installation of the system, process piping, electronic systems, percussion and sensors, yard upgrades, buildings, and perhaps even the cost of WCC, which, as indicated by Herrera-Rodriguez et al. [57] may constitute 6.5% of the FCE. Consequently, the FCE to plan a production process with 1000 m3/day of aptitude reaches USD 507,000.00 as describe in Table 4.   

  • Reviewer Comment:  Just equipment prices are presented without any background.  
  • Authors' Responses: The equipment’s price was selected according to several previous studies and from the quotations of some companies.
  • What about utilities, hot water, steam, heat. Just needed money is mentioned without any information on what and in which quantity is used.
  • Line 297: The comprises electricity and water that are mandatory for the operation progression and predictable grounded on the price for each component in the native exchange. The ???, characterize the ultimate biomass yield produced in the production process of the PhACs.
  • Reviewer Comment: What about the waste production of your technology?
  • Authors' Responses: (Line 342-344). In addition, the cost of waste production will be calculating when this techno economic analysis applies for a future study.   
  • Reviewer Comment: There is still plenty of missing information to find your TEA reliable).
  • Authors' Responses:
  • (Line 420). This study represents a preliminary study in the TEA which is an initial exploration of the TEA technology by using SuperPro Designer The complete information will be in the next future study which will calculate the amount of hot water, steam, heat and waste production. In addition, the flowrates of individual substances.

Two references were added

  1. Gopalakrishnan, Y., Al-Gheethi, A., Abdul Malek, M., Marisa Azlan, M., Al-Sahari, M., Radin Mohamed, R. M. S., Noman, E. Removal of Basic Brown 16 from Aqueous Solution Using Durian Shell Adsorbent, Optimisation and Techno-Economic Analysis. Sustainability. (2020). 12(21), 8928. https://doi.org/10.3390/su12218928.
  2. Jones, S. B., Zhu, Y., Anderson, D. B., Hallen, R. T., Elliott, D. C., Schmidt, A. J., Drennan, C. Process design and economics for the conversion of algal biomass to hydrocarbons: whole algae hydrothermal liquefaction and upgrading. (No. PNNL-23227). Pacific Northwest National Lab.(PNNL), Richland, WA (United States) (2014).
